# Adaptive fuzzy impedance control of human-robot interaction modular robot manipulators based on human motion intention estimation

1st Bo Dong
Department of Control Science and Engineering
Changchun University of Technology
Changchun, China
dongbo@ccut.edu.cn

2nd Rui Sun
Department of Control Science and Engineering
Changchun University of Technology
Changchun, China
sunruisr55@163.com

3rd Tianjiao An*
Department of Control Science and Engineering
Changchun University of Technology
Changchun, China
antianjiao@ccut.edu.cn

4th Chen Li
Department of Control Science and Engineering
Changchun University of Technology
Changchun, China
lichen163yx@163.com

5th Bing Ma
Department of Control Science and Engineering
Changchun University of Technology
Changchun, China
mabing@ccut.edu.cn

*Abstract*—For modular robot manipulators (MRMs) with physical human-robot interaction (pHRI) tasks, an adaptive fuzzy impedance control method based on the estimation of human motion intention is proposed in this paper. Dynamic subsystem of MRM is built based on joint torque feedback (JTF) technique. Processing unknown human limb models using fuzzy logic systems (FLSs), the obtained intention of the human is considered as the desired trajectory for impedance control, enabling the robot to actively follow a target impedance model. Adaptive fuzzy impedance control is utilized to compensate for model uncertainty and accomplish tracking goals. The uniformly ultimately bounded (UUB) of the tracking error is verified through Lyapunov theory. Eventually, the validity of the proposed adaptive fuzzy impedance control method is experimentally verified.

*Index Terms*—physical human-robot interaction (pHRI), fuzzy logic systems (FLSs), modular robot manipulator (MRM), impedance control

## I. Introduction

Compared with traditional robots, modular robot manipulators (MRMs) are reconfigurable and have good flexibility and versatility, which have an increasing influence in robotics. Since the first industrial MRM systems were designed and developed in the world, MRMs have rapidly developed and been applied [1]. Today, MRMs play a crucial role in areas such as post-disaster rescue, aerospace and physical human-robot interaction (pHRI).

In daily life, there are inevitably some tasks that require a minimum of two people to complete, such as moving larger objects can't be done by one person, and collaboration robot can assist human in accomplishing tasks. Robot can ensure the safety of the collaborator in human-robot collaboration tasks by obtaining the intention of the collaborator [2]. If the robot can acquire the intention of the human, it can use the result as its desired trajectory, then pHRI is realized [3]. Therefore, the robot recognition of human motion intention is the key to pHRI [4]. Using an electromyography-based method, Bednarczyk et al. [5] distinguished interactive forces from forces generated by interacting with the environment to capture the operator's motion intention. Due to the complexity of the human body, methods based on human biological signals for intention recognition have significant flaws. Compared to using human biological signals for intention recognition, utilizing robot's own sensor signals overcome human body uncertainties [6]. Li et al. [7] used the position and force information obtained from the sensors to apply a neural network for intention recognition. An et al. [8] proposed a harmonic drive compliance model-based intention recognition, which only required the position measurement of the robot.

In pHRI tasks, applying appropriate control strategies can ensure system stability. Impedance control is widely used

The work is supported by the National Natural Science Foundation of China (62173047), the Scientific Technological Development Plan Project in Jilin Province of China (20220201038GX), Key Laboratory of Advanced Structural Materials (Changchun University of Technology), Ministry of Education, China (ASM-202202).

in the pHRI due to its good robustness [9]. Fuzzy logic systems (FLSs) are considered to have ability to approximate nonlinear systems [10]. Zhang et al. [11] solved the target tracking control challenges in nonlinear pure-feedback systems by FLSs. Zhao et al. [12] used FLSs to accomplish the trajectory tracking control for a category of low triangular nonlinear systems with uncertainty. Wang et al. [13] used FLSs approximation of unknown nonlinear functions and ensured the trajectory tracking tasks are accomplished.

As a result of the above discussion, an adaptive fuzzy impedance control method for pHRI oriented MRM systems based on human motion intention estimation is proposed in this paper. First, the joint torque feedback (JTF) technique is used to establish the MRM system dynamic model. Then, an FLS is introduced, which is applied to recognize the human motion intention, and the recognized result is treated as the expected trajectory in the impedance model. An adaptive fuzzy impedance controller is used to compensate for model uncertainty so that the robot can successfully accomplish the position tracking tasks. Finally, Lyapunov theory demonstrates the stability of the MRM system, and the results of the experiment verify the effectiveness of the proposed method.

## II. ESTABLISHING THE DYNAMIC MODEL AND PREPARATION WORK

### A. Dynamic modeling analysis

In this paper, an $n$ degrees of freedom MRM has considered and the expression of the dynamic model for the $i$th joint subsystem in the MRM is given by:

$$I_{mi}\beta_i\ddot{\alpha}_i + f_{li}(\alpha_i, \dot{\alpha}_i) + L_i(\alpha, \dot{\alpha}, \ddot{\alpha}) + \frac{\tau_{ci}}{\beta_i} = \tau_i + \tau_{jfi} \quad (1)$$

where $\alpha$, $\dot{\alpha}$ and $\ddot{\alpha}$ are the position, velocity and acceleration vector of the $i$th joint of the modular robot. $\tau_i$ is the control torque. $\tau_{jfi}$ is pHRI torque. $\tau_{ci}$ represents the joint torque detected by the sensor. $I_{mi}$ means the rotor's moment of inertia about the axis of rotation. $\beta_i$ indicates the reduction ratio of the motor. $L_i(\alpha, \dot{\alpha}, \ddot{\alpha})$ denotes the interconnected dynamic coupling (IDC) terms. $f_{li}(\alpha_i, \dot{\alpha}_i)$ is the joint friction torque.

1) The IDC items:

$$L_i(\alpha, \dot{\alpha}, \ddot{\alpha}) = I_{mi}\sum_{n=1}^{i-1} h_{mi}^T h_{qn}\ddot{\alpha}_n$$
$$+ I_{mi}\sum_{n=2}^{i-1}\sum_{s=1}^{n-1} h_{mi}^T(h_{ls} \times h_{qn})\dot{\alpha}_s\dot{\alpha}_n$$
$$= I_{mi}\sum_{n=1}^{i-1} A_n^i\ddot{\alpha}_n + \ddot{I}_{mi}\sum_{n=2}^{i-1}\sum_{s=1}^{n-1} B_{sn}^i\dot{\alpha}_s\dot{\alpha}_n$$
$$= \sum_{n=1}^{i-1}\left[I_{mi}\hat{A}_n^i I_{mi}\right]\left[\ddot{\alpha}_n\tilde{A}_n^i\ddot{\alpha}_n\right]^T$$
$$+ \sum_{n=2}^{i-1}\sum_{s=1}^{n-1}\left[I_{mi}\hat{B}_{sn}^i I_{mi}\right]\left[\dot{\alpha}_s\dot{\alpha}_n\tilde{B}_{sn}^i\dot{\alpha}_s\dot{\alpha}_n\right]^T$$
$$= \sum_{n=1}^{i-1} U_n^i Q_n^i + \sum_{n=2}^{i-1}\sum_{s=1}^{n-1} S_{sn}^i R_{sn}^i. \quad (2)$$

The IDC items can be reexpressed as follows:

$$L_i(\alpha, \dot{\alpha}, \ddot{\alpha}) = \sum_{n=1}^{i-1} U_n^i Q_n^i + \sum_{n=2}^{i-1}\sum_{s=1}^{n-1} S_{sn}^i R_{sn}^i = Q_z^i + R_z^i \quad (3)$$

where $h_{mi}$ is the unit vector around the $i$th motor rotation axis, $h_{qn}$ and $h_{ls}$ are the unit vectors in the direction of the axis of rotation of the $n$th joint and the $s$th joint. $A_n^i = h_{mi}^T h_{qn}$ means dot product of $h_{mi}^T$, $h_{qn}$. $\tilde{A}_n^i$ represents the alignment error of $A_n^i$, $\tilde{A}_n^i + \hat{A}_n^i = A_n^i$. $B_{sn}^i = h_{mi}^T(h_{ls} \times h_{qn})$ denotes dot product of $h_{mi}^T$, $(h_{ls} \times h_{qn})$. $\tilde{B}_{sn}^i$ means the alignment error of $B_{sn}^i$, $\tilde{B}_{sn}^i + \hat{B}_{sn}^i = B_{sn}^i$.

2) The joint friction torque:

$$f_{li}(\alpha_i, \dot{\alpha}_i) = f_{\alpha i}(\alpha_i, \dot{\alpha}_i) + f_{bi}\dot{\alpha}_i$$
$$+ \left(f_{ci} + f_{si}e^{\left(-f_{ti}\dot{\alpha}_i^2\right)}\right)\text{sgn}(\dot{\alpha}_i) \quad (4)$$

where $f_{\alpha i}(\alpha_i, \dot{\alpha}_i)$ is the position-dependent friction parameter, $f_{bi}$ denotes the viscous friction parameters, $f_{ci}$ is the coulomb friction parameters, $f_{si}$ is the static friction, $f_{ti}$ is the Stribeck effect parameter. According to the proposed linearization strategy, one can obtain:

$$f_{si}e^{\left(-f_{ti}\dot{\alpha}_i^2\right)} \approx \hat{f}_{si}e^{\left(-\hat{f}_{ti}\dot{\alpha}_i^2\right)} + \tilde{f}_{si}e^{\left(-\hat{f}_{ti}\dot{\alpha}_i^2\right)}$$
$$- \dot{\alpha}_i^2\tilde{f}_{ti}\hat{f}_{si}e^{\left(-\hat{f}_{ti}\dot{\alpha}_i^2\right)}. \quad (5)$$

Substituting (5) into (4) yields:

$$\hat{f}_{li}(\alpha_i, \dot{\alpha}_i) = f_{\alpha i}(\alpha_i, \dot{\alpha}_i) + \hat{f}_{bi}\dot{\alpha}_i + Z(\dot{\alpha}_i)\tilde{F}$$
$$+ \left(\hat{f}_{ci} + \hat{f}_{si}e^{\left(-\hat{f}_{ti}\dot{\alpha}_i^2\right)}\right)\text{sgn}(\dot{\alpha}_i) \quad (6)$$

where $\tilde{F}_i = \left[-\hat{f}_{bi} + f_{bi}, -\hat{f}_{ci} + f_{ci}, -\hat{f}_{si} + f_{si}, -\hat{f}_{ti} + f_{ti}\right]^T$ denotes the uncertain friction parameters, $Z(\dot{\alpha}_i)$ is denoted as follows:

$$Z(\dot{\alpha}_i) = \left[\dot{\alpha}_i, \text{sgn}(\dot{\alpha}_i), e^{\left(-\hat{f}_{ti}\dot{\alpha}_i^2\right)}\text{sgn}(\dot{\alpha}_i),\right.$$
$$\left. -\hat{f}_{si}\dot{\alpha}_i^2 e^{\left(-\hat{f}_{ti}\dot{\alpha}_i^2\right)}\text{sgn}(\dot{\alpha}_i)\right]. \quad (7)$$

Substituting (3) and (6) into (1), one has:

$$\ddot{\alpha}_i = -N_i\left[\begin{array}{c} f_{\alpha i}(\alpha_i, \dot{\alpha}_i) + \hat{f}_{bi}\dot{\alpha}_i + Z(\dot{\alpha}_i)\tilde{F}_i \\ -\tau_i - \tau_{jfi} + \frac{\tau_{ci}}{\beta_i} + Q_z^i + R_z^i \\ + \left(\hat{f}_{ci} + \hat{f}_{si}e^{\left(-\hat{f}_{ti}\dot{\alpha}_i^2\right)}\right)\text{sgn}(\dot{\alpha}_i) \end{array}\right] \quad (8)$$

where $N_i = (I_{mi}\beta_i)^{-1}$, define state variables as $x_i = [x_{i1}\ x_{i2}]^T = [\alpha_i\ \dot{\alpha}_i]^T \in R^{2\times1}$, $u_i = \tau_i \in R^{1\times1}$ is the control input of the systems. The MRM subsystem state space can be represented as:

$$\begin{cases} \dot{x}_{i1} = x_{i2} \\ \dot{x}_{i2} = C_i(x_i) + N_i u_i + N_i\tau_{jfi} + T_i(x) \\ y_i = x_{i1}. \end{cases} \quad (9)$$

The precise modeling item and measurable part of the subsystem dynamic model is denoted as:

$$C_i(x_i) = -N_i\left[\begin{array}{c} \left(\hat{f}_{ci} + \hat{f}_{si}e^{\left(-\hat{f}_{ti}\dot{\alpha}_i^2\right)}\right)\text{sgn}(\dot{\alpha}_i) \\ + \frac{\tau_{ci}}{\beta_i} + \hat{f}_{bi}\dot{\alpha}_i \end{array}\right] \quad (10)$$

where $T_i(x) = -N_i \left[ Q_z^i + R_z^i + f_{\alpha i}(\alpha_i, \dot{\alpha}_i) + Z(\dot{\alpha}_i)\tilde{F}_i \right]$ represents the uncertainty component of the subsystem.

### B. Fuzzy Logic System Representation

FLSs have strong approximation capabilities and are often used as approximation nonlinear functions. The IF-THEN rules in FLSs can be defined as follows:

$$R_i^k : \text{If } x_{i1} \text{ is } F_{i1}^k, \; x_{i2} \text{ is } F_{i2}^k, \; ..., \; x_{in} \text{ is } F_{in}^k,$$
$$\text{Then } y_i \text{ is } H_i^k, \; k = 1, 2, \cdots, N \quad (11)$$

where $x_{ij}$ denotes the input of FLSs, $y_i$ denotes the output of FLSs, $F_{ij}^k$ and $H_i^k$ represent the fuzzy sets. One can obtain:

$$y_i = \frac{\sum_{k=1}^{N} \psi_k \prod_{j=1}^{n} \zeta_{F_{ij}^k}}{\sum_{k=1}^{N} \left( \prod_{j=1}^{n} \zeta_{F_{ij}^k} \right)} \quad (12)$$

where $\psi_k = \max \zeta_{H_i^k}$. Define the fuzzy basis function as follows:

$$\zeta_k = \frac{\prod_{j=1}^{n} \zeta_{F_i^k}}{\sum_{k=1}^{N} \left( \prod_{j=1}^{n} \zeta_{F_i^k} \right)}. \quad (13)$$

Equation (12) can be redescribed as follows:

$$y_i = W_i^T \zeta_j \quad (14)$$

where $W = [W_1, \ldots, W_N]^T = [\psi_1, \ldots, \psi_N]^T$.

**Lemma 1** [14]: For a given continuous function $y_i$, there is an FLS which satisfies $\sup_{x_i} \left| y_i - W_i^T \zeta \right| \leq \eta_i$, $\eta_i$ denotes an arbitrarily positive constant.

## III. ADAPTIVE FUZZY IMPEDANCE CONTROLLER DESIGN BASED ON HUMAN MOTION INTENTION ESTIMATION

### A. Estimation of human motion intention

In pHRI, a human limb model in the Cartesian space is defined as:

$$-C_H \dot{x} + G_H(x_{Hd} - x) = f \quad (15)$$

where $C_H$ is the human's damper matrix, $G_H$ is the human's spring matrix, $x_{Hd}$ denotes the desired trajectory of the human, $f$ represents interaction force, the motion intention $x_{Hd}$ is expressed as:

$$x_{Hd} = Y(f, x, \dot{x}) \quad (16)$$

where $Y(\cdot)$ represents an unknown function, since $Y(\cdot)$ is an unknown nonlinear function, approximating it by fuzzy logic.

The desired trajectory of the human and its estimation are denoted by:

$$x_{Hd,i} = \hat{W}_{1i}^T \zeta_{1i}(b_i) + \varepsilon_{1i}$$
$$\hat{x}_{Hd,i} = \hat{W}_{1i}^T \zeta_{1i}(b_i), \; i = 1, 2, \ldots, n \quad (17)$$

where $b_i = \left[ f_i^T, \; x_i^T, \; \dot{x}_i^T \right]^T$ represents the input of FLSs. Using the steepest descent method, one can obtain:

$$E_i = \frac{1}{2} f_i^2. \quad (18)$$

Then,

$$\begin{aligned}
\dot{\hat{W}}_{1i}(t) &= -a_i' \frac{\partial E_i}{\partial \hat{W}_{1i}} \\
&= -a_i' \frac{\partial E_i}{\partial f_i} \frac{\partial f_i}{\partial x_{Hd,i}} \frac{\partial x_{Hd,i}}{\partial \hat{W}_{1i}}
\end{aligned} \quad (19)$$

where $\frac{\partial f_i}{\partial x_{Hd,i}}$, $\frac{\partial x_{Hd,i}}{\partial \hat{W}_{1i}}$ are obtained by:

$$\frac{\partial f_i}{\partial x_{Hd,i}} = G_{H,i}, \; \frac{\partial x_{Hd,i}}{\partial \hat{W}_{1i}} = \zeta_{1i}(b_i). \quad (20)$$

The update adaptive rate can be designed as follows:

$$\dot{\hat{W}}_{1i}(t) = -a_i f_i \zeta_{1i}(b_i) \quad (21)$$

where $a_i = a_i' G_{H,i}$, $a_i'$ is a small positive scalar, $a_i'$ and $G_{H,i}$ can be absorbed by $a_i$. Eventually, the method to derive the update law of $\hat{W}_{1i}$ is:

$$\hat{W}_{1i}(t) = \hat{W}_{1i}(0) - a_i \int_0^t [f_i(w) \zeta_{1i}(b_i(w))] dw. \quad (22)$$

The estimation of human motion intention $\hat{x}_{Hd}$ can be obtained by substituting (22) into (17).

### B. Adaptive impedance control

The target impedance model of the MRM can be represented as follows:

$$M_{di}(\ddot{\alpha}_{di} - \ddot{\alpha}_i) + C_{di}(\dot{\alpha}_{di} - \dot{\alpha}_i) + G_{di}(\alpha_{di} - \alpha_i) = J_i^T f \quad (23)$$

where $M_{di}$, $C_{di}$ and $G_{di}$ are the desired matrices of inertia, damper and stiffness.

Since the human expected trajectory obtained earlier is in Cartesian space, in this section, it is transformed into joint space as:

$$\begin{aligned}
x &= \chi(\alpha), \dot{x} = J(\alpha)\alpha, \\
\ddot{x} &= \dot{J}(\alpha)\dot{\alpha} + J(\alpha)\ddot{\alpha}, \; \hat{\alpha}(t) = \chi^{-1}(\hat{x}_{Hd})
\end{aligned} \quad (24)$$

where $x$, $\dot{x}$, $\ddot{x} \in R^m$ represent position, velocity, acceleration vectors in the Cartesian space, $\chi(\cdot)$ represents the robot kinematics. Motion intention estimation $\hat{\alpha}_i$ is defined as the expected trajectory $\alpha_{di}$ and constructs error signal as follows:

$$\begin{aligned}
d &= M_{di}(\ddot{\alpha}_{di} - \ddot{\alpha}_i) + C_{di}(\dot{\alpha}_{di} - \dot{\alpha}_i) \\
&+ G_{di}(\alpha_{di} - \alpha_i) - J_i^T f.
\end{aligned} \quad (25)$$

Define the error signal as follows:

$$\begin{aligned}
e_i &= \alpha_{di} - \alpha_i, \\
\dot{e}_i &= \dot{\alpha}_{di} - \dot{\alpha}_i, \\
\ddot{e}_i &= \ddot{\alpha}_{di} - \ddot{\alpha}_i, J^T f = \tau_{jfi}.
\end{aligned} \quad (26)$$

To facilitate analysis, define another error signal as follows:

$$\bar{d} = K_{ai} d = \ddot{e}_i + K_{bi} e_i + K_{ci} e_i - K_{ai} \tau_{jfi} \quad (27)$$

where $K_{ai} = M_{di}^{-1}$, $K_{bi} = M_{di}^{-1}C_{di}$, $K_{ci} = M_{di}^{-1}G_{di}$.

Select two positive definite matrices $g_i$ and $h_i$ as follows:

$$g_i + h_i = K_{bi}, \quad \dot{g}_i + h_i g_i = K_{ci}. \tag{28}$$

Defining $K_{ai}\tau_{jfi} = \dot{\tau}_{jfli} + \tau_{jfli}h_i$ yields:

$$\bar{d} = \ddot{e}_i + (g_i + h_i)e_i + (\dot{g}_i + h_i g_i)e_i - \dot{\tau}_{jfi} - \tau_{jfi}h_i. \tag{29}$$

Define an auxiliary variable $z$ as:

$$z = \dot{e}_i + g_i e_i - \tau_{jfli}. \tag{30}$$

One can obtain:

$$\bar{d} = \dot{z}_i + h_i z. \tag{31}$$

Suppose that the existence of $\lim_{t\to\infty} z(t) = 0$ will make $\lim_{t\to\infty} \dot{z}(t) = 0$, $\lim_{t\to\infty} d(t) = 0$, then the control objective $\lim_{t\to\infty} z(t) = 0$ is achieved.

Differentiating equation (30) yields:

$$\ddot{e}_i = \dot{z} - \dot{g}_i e_i - g_i \ddot{e}_i + \dot{\tau}_{jfli}. \tag{32}$$

Substituting (26) and (32) into (7) leads to:

$$C_i(x_i) + N_i u_i + N_i \tau_{jf} + T_i(x)$$
$$= \ddot{\alpha}_{di} - \dot{z} + \dot{g}_i e_i + g_i \dot{e}_i + g_i \ddot{e}_i - \dot{\tau}_{jfli}$$
$$I_{mi}\beta_i C_i(x_i) + u_i + \tau_{jfi} + I_{mi}\beta_i \dot{z} \tag{33}$$
$$= I_{mi}\beta_i \left( \begin{array}{c} \ddot{\alpha}_{di} - T_i(x) + \dot{g}_i e_i \\ + g_i \dot{e}_i + g_i \ddot{e}_i - \dot{\tau}_{jfli} \end{array} \right).$$

By defining $H(\Phi) = I_{mi}\beta_i(\ddot{\alpha}_{di} - T_i(x) + \dot{g}_i e_i + g_i \dot{e}_i + g_i \ddot{e}_i - \dot{\tau}_{jfli})$, one can obtain:

$$u_i = -I_{mi}\beta_i C_i(x_i) - \tau_{jfi} - I_{mi}\beta_i \dot{z} + H(\Phi). \tag{34}$$

Adaptive fuzzy impedance controller can be designed as:

$$u_i = -I_{mi}\beta_i C_i(x_i) - \tau_{jfi} + \hat{H}(\Phi) - \varepsilon_{2i} + K_{g1}z$$
$$= u_{i1} - \tau_{jfi} - I_{mi}\beta_i C_i(x_i) + K_{g1}z \tag{35}$$

where $K_{g1}$ is the positive-definite matrix, $u_{i1} = \hat{W}_{2i}^T S_1(\Phi) = \hat{H}(\Phi) - \varepsilon_{2i}$. The adaptive law is as follows:

$$\dot{\hat{W}}_{2i} = \Lambda_i[\hat{W}_{2i}^T S_{1i}(\Phi) - \delta_i \hat{W}_{2i}], \quad i = 1, 2, \ldots, n. \tag{36}$$

Substituting (35) into (33), one has:

$$I_{mi}\beta_i \dot{z} = -K_{g1}z - \tilde{W}_{2i}^T S_1(\Phi) + \varepsilon_{2i} \tag{37}$$

where $\tilde{W}_{2i}$ denotes the weight error, $\tilde{W}_{2i} = \hat{W}_{2i} - W_{2i}$.

**Theorem 1.** Considering the dynamic model of the MRM system in (1) and its state space representation in (9), the trajectory tracking error under the pHRI task is uniformly ultimately bounded (UUB) via the proposed control method in (35).

**Proof:** Choose the following Lyapunov candidate functions as bellow:

$$V = \sum_{i=1}^{n} \left( \frac{1}{2} z^T N_i z + \frac{1}{2} \tilde{W}_{2i}^T \Lambda_i^{-1} \tilde{W}_{2i} \right). \tag{38}$$

Derivation for $V$ with respect to time, one can obtain:

$$\dot{V} = \sum_{i=1}^{n} \left( z^T N_i \dot{z} + \tilde{W}_{2i}^T \Lambda_i^{-1} \dot{\hat{W}}_{2i} \right). \tag{39}$$

Substituting (36) and (37) into (39) leads to:

$$\dot{V} = z^T \left[ -K_{g1}z - \tilde{W}_{2i}^T S_1(\Phi) + \varepsilon_{2i} \right] + \tilde{W}_{2i}^T \Lambda_i^{-1} \dot{\hat{W}}_{2i}$$
$$= -z^T K_{g1}z + z^T \varepsilon_{2i} + \tilde{W}_{2i}^T \left[ \Lambda_i^{-1} \dot{\hat{W}}_{2i} - z^T S_1(\Phi) \right]. \tag{40}$$

Substituting the adaptive law (36) into (40), one can obtain:

$$\dot{V} = -z^T K_{g1}z + z^T \varepsilon_{2i} - z^T S_1(\Phi)$$
$$+ \tilde{W}_{2i}^T \left\{ \Lambda_i^{-1} \left[ \Lambda_i \left( \hat{W}_{2i}^T S_{1i}(\Phi) - \delta_i \hat{W}_{2i} \right) \right] \right\} \tag{41}$$
$$= -z^T K_{g1}z + z^T \varepsilon_{2i} - \tilde{W}_{2i}^T \delta_i W_{2i} - \tilde{W}_{2i}^T \delta_i \tilde{W}_{2i}.$$

Then, we have:

$$\dot{V} \leq -z^T (K_{g1} - \frac{1}{2}I_{n\times n})z + \frac{1}{2}\|\varepsilon_{2i}(\Phi)\|^2$$
$$+ \frac{\delta_i}{2}(\|W_{2i}\|^2 - \|\tilde{W}_{2i}\|^2) \tag{42}$$
$$\leq -\rho V_2 + C$$

where

$$\rho = \min\left( \min\left( \frac{2\zeta_{\min}(K_{g1} - \frac{1}{2}I)}{\zeta_{\max}(N_i)}, \min\left( \frac{\delta_i}{\zeta_{\max}(\Lambda_i^{-1})} \right) \right) \right)$$
$$C = \frac{1}{2}\|\bar{\varepsilon}_{2i}\|^2 + \sum_{i=1}^{n} \frac{\delta_i}{2}\|W_{2i}\|^2 \tag{43}$$

where $\zeta$ is matrix eigenvalue. In order to make $\rho > 0$, it is necessary to make $\zeta_{\min}(K_{g1} - \frac{1}{2}I) > 0$, $\zeta_{\max}(\Lambda_i^{-1}) > 0$, $C$ is a constant, $\bar{\varepsilon}_{2i}$ is the boundary of $\varepsilon_{2i}$.

$$\Omega_z = \left\{ z R^n | \|z\| \leq \sqrt{\frac{K_e}{\zeta_{\min}(N_i)}} \right\}$$
$$\Omega_w = \left\{ \tilde{W} R^{l\times n} | \|\tilde{W}\| \leq \sqrt{\frac{K_e}{\zeta_{\min}(\Lambda^{-1})}} \right\} \tag{44}$$

where $K_e = 2(C + V(0))/\rho$.

## IV. EXPERIMENT

The effectiveness of the proposed method is validated on a two degrees of freedom MRM experimental platform in this part. For the proposed adaptive fuzzy impedance control method, the inputs to the fuzzy logic are position error $e_i$ and velocity error $\dot{e}_i$, the output to the fuzzy logic is $u_{i1}$. Then, the rule base consists of $5^m$ rules, as shown in Table 1. Fig. 1 and Fig. 2 represent the position tracking curves. Fig. 3 represents the tracking error curves for the two joints, and it shows that the robot can accomplish the tracking tasks. Fig. 4 represents the control torque curves. Fig. 5 represents the fuzzy logic weights in the proposed adaptive fuzzy impedance control method. Fig. 6 shows the fuzzy membership functions. From the above experiments, it can be summarized that the interaction tasks can be successfully accomplished by using the proposed method.

TABLE I
FUZZY LOGIC CONTROLLER RULE BASE.

| The input of fuzzy logic | | Velocity tracing error | | | | |
|---|---|---|---|---|---|---|
| | | Positive (P) | Positive medium (Pm) | Zero (Z) | Negative medium (Nm) | Negative (N) |
| Position tracking error | Positive (P) | P | P | Pm | Pm | Z |
| | Positive medium (Pm) | P | Pm | Pm | Z | Nm |
| | Zero (Z) | Pm | Pm | Z | Nm | Nm |
| | Negative medium (Nm) | Pm | Z | Nm | Nm | N |
| | Negative (N) | Z | Nm | Nm | N | N |

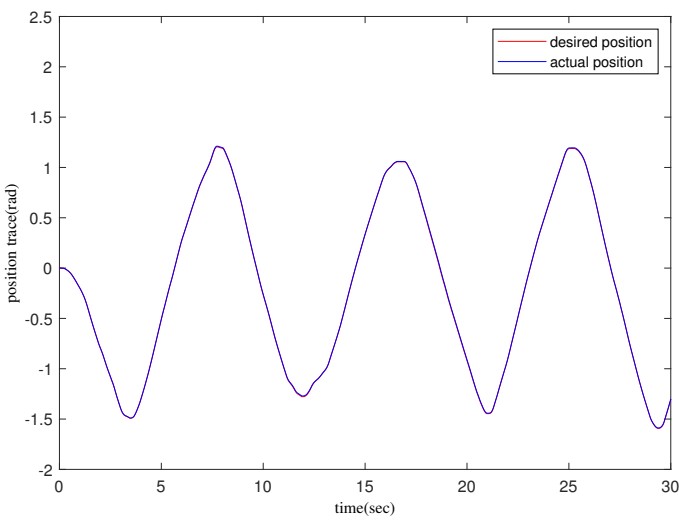

Fig. 1. Joint 1 position tracking in joint space via the proposed adaptive fuzzy impedance control method.

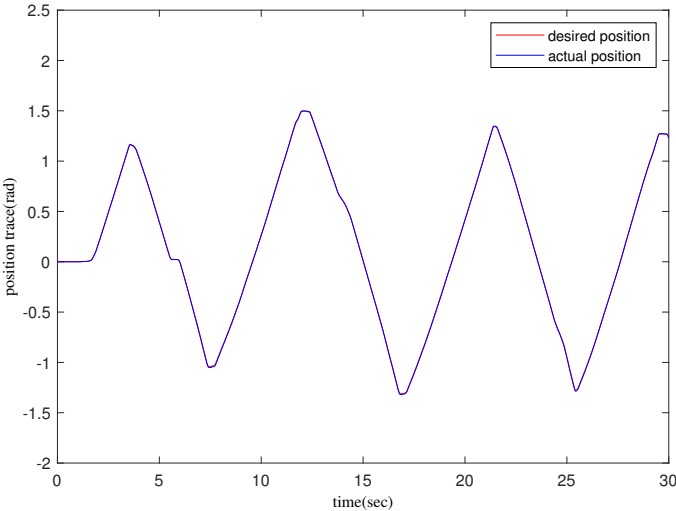

Fig. 2. Joint 2 position tracking in joint space generated by the proposed adaptive fuzzy impedance control method.

Fig. 3. Curves of the position tracking errors in joint space by the proposed adaptive fuzzy impedance control method. (a) Joint1, (b) Joint2.

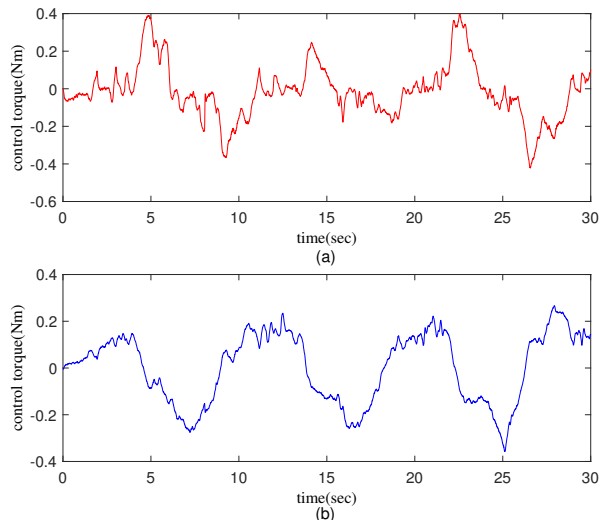

Fig. 4. Control torque curves generated by the proposed adaptive fuzzy impedance control method. (a) Joint1, (b) Joint2.

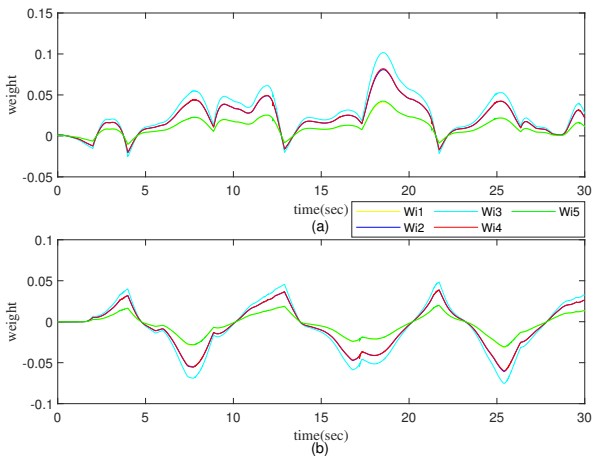

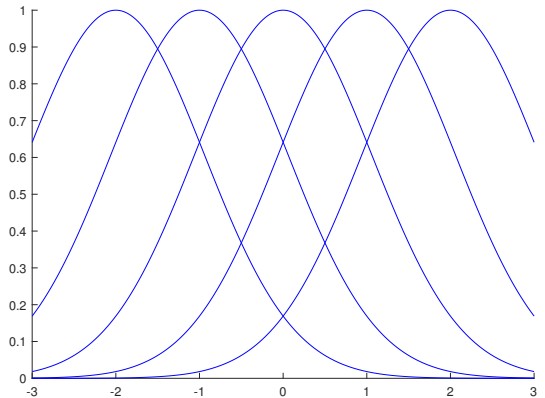

Fig. 5. Fuzzy logic weight vector curves obtained by the proposed adaptive fuzzy impedance control method. (a) Joint1, (b) Joint2.

Fig. 6. Fuzzy membership function curves.

## V. CONCLUSION

This paper introduces an adaptive fuzzy impedance control approach based on intent recognition. Based on the establishment of the MRM dynamic model, we use FLSs for intent recognition, and let the obtained result as the desired trajectory for impedance control. Then, an adaptive fuzzy impedance controller to compensate for the model uncertainty while completing the tracking of the target is designed. MRM system is demonstrated to be UUB via Lyapunov theory. Verification of the validity of the proposed adaptive fuzzy impedance control method is accomplished by experiment.

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
