# OpenReview forum: "Adaptive fuzzy impedance control of human-robot interaction modular robot manipulators based on human motion intention estimation"
_IEEE.org/ICIST/2024/Conference — IEEE ICIST 2024 Conference Submission_

### Official Review · Reviewer_7YLS · 2024-08-21
**accept**

**Rating:** 7
**Confidence:** 3

**Review:**

Comment: This paper proposes an adaptive fuzzy impedance control method based on the estimation of human motion intention for modular robot manipulators (MRMs) with physical human-robot interaction (pHRI) tasks. The theory is correct and can be accepted after responding the following comments.
(1) More comprehensive literature review is needed to clarify the research gap and research motivation.
(2) In the simulation section, more analysis can be added to better explain the main results of this paper, that's not enough.
(3) In the end of the conclusions, some research directions are suggested to be added.

---

### Official Review · Reviewer_fhrg · 2024-08-22
**Accept**

**Rating:** 7
**Confidence:** 3

**Review:**

In this paper, " Adaptive fuzzy impedance control of human-robot interaction modular robot manipulators based on human motion intention estimation", an adaptive fuzzy impedance control method based on human motion intention estimation is proposed. Firstly, the dynamic subsystem of MRM is established based on joint torque feedback (JTF) technology. Then, the fuzzy logic system (FLSs) is used to process the unknown human limb model. The article has clear logic and organization, but there are still some problems. My specific feedback is as follows :1) In the introduction part, the author's analysis of fuzzy impedance control is insufficient. 2) What are the advantages of the PHRI-based adaptive fuzzy impedance control method for MRM system compared with the traditional method?

---

### Official Review · Reviewer_nXGu · 2024-08-22
**This article is very interesting and a good one**

**Rating:** 7
**Confidence:** 3

**Review:**

In this paper, an adaptive fuzzy impedance control method based on the estimation of human motion intention was proposed for MRMs with pHRI tasks. The obtained result is valuable and can be accepted if the following problems can be clarified.
(1) In the introduction, the shortages of those relevant studies are suggested to be further summarized.
(2) In the end of Section 1, the organization of this study is suggested to be summarized.
 (3) There exist several spelling and grammar errors. Please check carefully and further polish
(4) In the simulation section, more analysis can be added to better explain the main results of this paper, that's not enough.
(5) The future work is missing in the Conclusion.

---

### Comment · Reviewer_nXGu · 2024-08-21
**This article is very interesting and a good one**

In this paper, an adaptive fuzzy impedance control method based on the estimation of human motion intention was proposed for MRMs with pHRI tasks. The obtained result is valuable and can be accepted if the following problems can be clarified.
(1)	In the introduction, the shortages of those relevant studies are suggested to be further summarized.
(2)	In the end of Section 1, the organization of this study is suggested to be summarized.
(3)	There exist several spelling and grammar errors. Please check carefully and further polish
(4)	In the simulation section, more analysis can be added to better explain the main results of this paper, that's not enough.
(5)	The future work is missing in the Conclusion.

---

### Decision · Program_Chairs · 2024-09-06

Accept (Oral)